# An Algoclay-Based Decontaminant Decreases Exposure to Aflatoxin B_1_, Ochratoxin A, and Deoxynivalenol in a Toxicokinetic Model, as well as Supports Intestinal Morphology, and Decreases Liver Oxidative Stress in Broiler Chickens Fed a Diet Naturally Contaminated with Deoxynivalenol

**DOI:** 10.3390/toxins16050207

**Published:** 2024-04-26

**Authors:** Marie Gallissot, Maria A. Rodriguez, Mathias Devreese, Isis Van herteryck, Francesc Molist, Regiane R. Santos

**Affiliations:** 1Olmix SA, Z.A. du Haut du Bois, 56580 Brehan, France; mrodriguez@olmix.com; 2Department of Pathobiology, Pharmacology and Zoological Medicine, Faculty of Veterinary Medicine, Ghent University, Salisburylaan 133, 9820 Merelbeke, Belgium; mathias.devreese@ugent.be (M.D.); isis.vanherteryck@ugent.be (I.V.h.); 3Department of Research and Development, Schothorst Feed Research, Meerkoetenweg 26, 8218 NA Lelystad, The Netherlands; fmolist@schothorst.nl (F.M.); rsantos@schothorst.nl (R.R.S.)

**Keywords:** *Fusarium*, mycotoxins, algae, intestine, liver, production performance, broiler chickens

## Abstract

The aims of this study were (i) to determine the effect of an algoclay-based decontaminant on the oral availability of three mycotoxins (deoxynivalenol; DON, ochratoxin A; OTA, and aflatoxin B_1_; AFB_1_) using an oral bolus model and (ii) to determine the effect of this decontaminant on the performance, intestinal morphology, liver oxidative stress, and metabolism, in broiler chickens fed a diet naturally contaminated with DON. In experiment 1, sixteen 27-day-old male chickens (approximately 1.6 kg body weight; BW) were fasted for 12 h and then given a bolus containing either the mycotoxins (0.5 mg DON/kg BW, 0.25 mg OTA/kg BW, and 2.0 mg AFB_1_/kg BW) alone (*n* = 8) or combined with the decontaminant (2.5 g decontaminant/kg feed; circa 240 mg/kg BW) (*n* = 8). Blood samples were taken between 0 h (before bolus administration) and 24 h post-administration for DON-3-sulphate, OTA, and AFB_1_ quantification in plasma. The algoclay decontaminant decreased the relative oral bioavailability of DON (39.9%), OTA (44.3%), and AFB_1_ (64.1%). In experiment 2, one-day-old male Ross broilers (*n* = 600) were divided into three treatments with ten replicates. Each replicate was a pen with 20 birds. The broiler chickens were fed a control diet with negligible levels of DON (0.19–0.25 mg/kg) or diets naturally contaminated with moderate levels of DON (2.60–2.91 mg/kg), either supplemented or not with an algoclay-based decontaminant (2 g/kg diet). Jejunum villus damage was observed on day 28, followed by villus shortening on d37 in broiler chickens fed the DON-contaminated diet. This negative effect was not observed when the DON-contaminated diet was supplemented with the algoclay-based decontaminant. On d37, the mRNA expression of glutathione synthetase was significantly increased in the liver of broiler chickens fed the DON-contaminated diet. However, its expression was similar to the control when the birds were fed the DON-contaminated diet supplemented with the algoclay-based decontaminant. In conclusion, the algoclay-based decontaminant reduced the systemic exposure of broiler chickens to DON, OTA, and AFB_1_ in a single oral bolus model. This can be attributed to the binding of the mycotoxins in the gastrointestinal tract. Moreover, dietary contamination with DON at levels between 2.69 and 2.91 mg/kg did not impair production performance but had a negative impact on broiler chicken intestinal morphology and the liver redox system. When the algoclay-based decontaminant was added to the diet, the harm caused by DON was no longer observed. This correlates with the results obtained in the toxicokinetic assay and can be attributed to a decreased absorption of DON.

## 1. Introduction

The composition and quality of feed have a crucial role in the production performance, health, and welfare of poultry. In addition to the advantageous elements included in a well-balanced diet, livestock feed frequently contains undesirable substances, such as toxins. The detrimental effects on the intestinal health of broiler chickens are well documented in relation to the mycotoxins generated by fungi belonging to the *Fusarium* species [1]. The authors of this study demonstrated that when broiler chickens were exposed to a concentration of 7.5 mg/kg of deoxynivalenol (DON) it led to compromised intestinal morphology. Additionally, this exposure generated oxidative stress and inflammation in both the gut and liver of the chickens. Nevertheless, it is important to note that detrimental effects on both production performance and intestinal morphology can occur even at mycotoxin levels that are below the recommended threshold of 5 mg/kg DON, as stated by the European Commission [2]. An illustration of this may be seen in studies where broiler chickens that were fed a maize-based diet containing 4 mg/kg DON exhibited intestinal and liver lesions together with a decline in performance [3]. If the birds are fed a wheat-based diet high in non-starch polysaccharides, a similar negative effect is observed at a much lower DON level (2.3 mg/kg) [4].

A newly published opinion by the European Food Safety Authority (EFSA) has shown evidence that dietary exposure to a concentration of 1.9 mg/kg has detrimental effects on bird intestinal morphology [5]. This is evident through the notable reduction in villus height in the jejunum. Depending on the severity of the damage, morphological alterations are also characterized by injury from the villus tip all the way to the crypt. In order to assess the extent of this particular form of harm, a scoring system is employed, wherein a higher score corresponds to a greater degree of intestinal injury [6]. Villus repair takes place spontaneously under normal physiological circumstances. Nevertheless, as a result of the cytotoxic impact of deoxynivalenol (DON), the process of protein synthesis is hindered, leading to a decline in the proliferation of intestinal epithelial cells and further villus recovery [7]. This decline in cell proliferation has the potential to impact the dimensions of villus height, crypt depth, and the overall healing mechanism of the intestinal tissue.

Deoxynivalenol also affects the redox balance [1] and metabolism [8] in the liver. Therefore, interventions to counteract the negative impact of this mycotoxin should also support gut and liver function. In laying hens, exposure to 10 mg/kg DON resulted in a short-term upregulation of glutathione synthetase (GSS) expression [9]. This upregulation has been considered a compensatory response to oxidative stress [10]. Depending on the severity of the oxidative stress, DON may stimulate an increase in the expression of inducible nitric oxide synthase (iNOS) [11]. In addition to oxidative stress, exposure to DON may also affect liver metabolism. Other mycotoxins, like aflatoxin B_1_, induce the downregulation of genes associated with fatty acid metabolism and energy production, such as carnitine palmitoyl transferase (CPT1) [12], which plays a role in the transport of long-chain fatty acids into mitochondria [13]. CPT1 is activated when a lack of energy is taking place [14]. Considering that DON causes energy failure and modulates β-oxidation [15], it may also regulate CPT1 expression. When exposed to DON, mammals’ 3-hydroxy-3-methylglutaryl-coenzyme A reductase (HMGCR) can temporarily increase the synthesis of lipids and cholesterol in the liver [16]. Broiler chickens subjected to chronic stress presented an upregulation in the expression of HMGCR in the liver, accompanied by hepatic cholesterol accumulation [17].

Multi-mycotoxin contamination of animal feed is unavoidable due to the feed composition, which consists of a variety of grains and cereals. To minimize animal exposure to these mycotoxins, it is suggested that dietary supplementation with decontaminants is performed to prevent the absorption of these toxins in the animal gastrointestinal tract. Bentonites are commonly used as mycotoxin binders and are registered in the European Union (EU) for their capacity to bind aflatoxin B_1_ (AFB_1_) [18]. Bentonites are phyllosilicates with variable physicochemical properties [19]. According to earlier research, alkyl groups may increase the bentonite interlayer space, which could enhance mycotoxin adsorption [20]. Nonetheless, Europe forbids the use of alkyl groups in animal feed. Alternatively, the bentonite interlayer gap can be increased by the polysaccharide ulvan, which is found in the cell wall of green seaweed (*Chlorophyceae*) belonging to the genus *Ulva* [21,22].

There is a scarcity of information on the efficacy of mycotoxin decontaminants on the in vivo absorption of mycotoxins. The EFSA stipulates that in vivo testing of mycotoxin decontaminants is necessary to evaluate the efficacy of the decontaminant. However, in vivo studies where non-specific parameters (e.g., growth, feed conversion, etc.) are measured are not sufficient for proving the efficacy of these products [23,24]. Toxicokinetic (TK) studies, based on the analysis of biomarkers for exposure are mandatory to determine the effects of decontaminants on the absorption of the target mycotoxin. These biomarkers can be the mycotoxin itself, but also a phase I or phase II metabolite. AFB_1_ and OTA can be measured directly in the plasma of broiler chickens [25]. For DON, only low concentrations are quantifiable in the plasma of broiler chickens, and DON-3-sulphate (DON-3S) has been shown to be the most relevant biomarker for exposure to DON in poultry [26].

The present study consisted of two experiments. In experiment 1, we aimed to determine the effect of an algoclay-based decontaminant on the oral availability of three mycotoxins (DON, OTA, and AFB_1_) using an oral bolus model. The plasma concentration of the relevant biomarkers for exposure to these mycotoxins was measured via UHPLC-MS/MS from 0 h (before administration) to 24 h post-administration (p.a.) of the bolus. In experiment 2, broiler chickens were fed a naturally contaminated diet containing moderate levels of DON (~3 mg/kg), which was the major contaminant in the diet. For this, broiler chickens were fed a marginally contaminated diet (control) or naturally DON-contaminated diets, either supplemented or not with an algoclay-based decontaminant at an inclusion of 2 g/kg. Production performance was assessed on days 14, 28, and 37. Intestinal morphology was assessed via histological analysis, where villus height, crypt depth, and villus were measured; damage in the intestinal villi was also scored. The liver tissue of the broiler chickens was subjected to qRT-PCR analyses for the analysis of the expression of GSS, iNOS, CPT1, and HMGC.

## 2. Results

### 2.1. Experiment 1: Toxicokinetic Assay

Ross broiler chickens (*n* = 16) were fed a commercial diet containing negligible levels of AFB_1_ (<0.02 mg/kg) and DON (<0.9 mg/kg), and no OTA was present before the toxicokinetic trial. On day 27, the broiler chickens weighed circa 1.6 kg BW (ranging from 1.5 to 1.9 kg BW) and were fasted for 12 h to guarantee that all birds had plasma levels of mycotoxins below the limit of quantification (LOQ, 0.5 ng/mL) at time point zero.

AFB_1_ was detected in the plasma of broiler chickens from 0.08 h up to 8 h p.a. (Figure 1a). The AUC_0→8h_ was significantly reduced when using the algoclay-based decontaminant, demonstrating a reduced systemic exposure to AFB_1_ via the decontaminant. The maximal plasma concentration (C_max_) was also reduced using the algoclay-based decontaminant. Furthermore, this C_max_ was reached later when using the decontaminant (lower T_max_) (Table 1).

The mycotoxin OTA was detected in the plasma of broiler chickens from 0.08 h up to 24 h p.a. (Figure 1b). The algoclay-based decontaminant significantly lowered the systemic exposure to OTA, with a decreased AUC_0→24h_ and a reduction in the relative oral bioavailability of OTA to 55.7% (Table 2).

DON-3S was detected in the plasma of broiler chickens from 0.08 h up to 12 h p.a. (Figure 1c). The algoclay-based decontaminant significantly lowered the systemic exposure to DON-3S, i.e., resulting in a decreased area under the curve (AUC)_0→12h_. The relative oral bioavailability of DON-3S in combination with the decontaminant was 60.1%, indicating the reduced exposure of DON-3S with the mycotoxin decontaminant (Table 3).

### 2.2. Experiment 2: Naturally Contaminated Diets

#### 2.2.1. Growth Performance

During the starter feeding period (d0–14), no differences were observed in feed intake (FI) and body weight gain (BWG). The feed conversion ratio (FCR) was significantly decreased in the broiler chickens fed the DON-contaminated diet (DON) compared to the control group, regardless of the presence of the algoclay-based decontaminant. During the grower feeding period (d14–28), BWG was significantly increased and FCR was decreased in the broiler chickens fed the DON diet, regardless of the presence of the algoclay-based test products. No differences in production performance were observed in the finisher (d28–37) and overall (d0–37) feeding periods. Furthermore, the mortality rate was not affected by the dietary treatments (Table 4).

#### 2.2.2. Intestinal Analysis

Although no significant differences in villus morphometry were observed on days 14 and 28, on d37, the jejunum villus height (VH) of broiler chickens fed the DON-contaminated diet significantly decreased. When the algoclay-based decontaminant was added to the DON diet, the VH was similar to that of the control diet. Also, on d37, a significant decrease in the villus height: crypt depth ratio (VH:CD) was observed in the jejunum of broiler chickens fed the DON-contaminated diet and this parameter was similar to the control when the algoclay-based decontaminant was added to the DON contaminated diet. Jejunum morphologic scores were not affected during the first 14 days of the trial. However, during the grower period, a significant increase in damage in the villus tip (score 1) was observed when the broiler chickens were fed the DON diet. This effect was not observed if the broiler chickens were fed the DON diet supplemented with the algoclay-based decontaminant. During the finisher feeding period, the intestinal damage scores were below 1 regardless of the treatment (Table 5). Representative images of the intestinal sections are given in Figure 2.

#### 2.2.3. mRNA Expression of Markers for Oxidative Stress and Metabolism in the Liver

Results for mRNA expression in liver tissue samples on days 14, 28, and 37 are given in Table 6. The expression of HMGCR was too low in all treatments. Therefore, it was not possible to compare the data. No differences in the expression of iNOS and CPT1 were observed, regardless of the sampling day. On day 37, liver mRNA GSS expression was significantly increased when the broiler chickens were fed the DON-contaminated diet, but supplementation of the contaminated diet with the algoclay-based decontaminant resulted in a GSS expression similar to the control group.

#### 2.2.4. DON and DON-3 Sulphate in the Serum

Serum levels of DON and DON-3S were determined at a single time-point on days 14, 28, and 37. In all samples, the DON serum levels were below the limit of quantification (LOQ), except for two samples, i.e., a 14-days-old bird fed the algoclay-supplemented DON diet (11.6 µg/mL DON) and a 28-days-old bird fed the non-supplemented DON diet (15.3 µg/mL DON). No significant differences in DON-3S could be observed, ranging from 0.75 to 1.47 ng/mL serum.

## 3. Discussion

In the present study, two experiments were conducted. In experiment 1, a TK study was conducted to determine the effect of an algoclay-based decontaminant on DON, OTA, and AFB_1_ oral bioavailability; in experiment 2, we evaluated the effect of a diet naturally contaminated with 2.66–2.91 mg/kg DON, either supplemented or not with an algoclay-based decontaminant, on growth performance, and intestinal morphology, as well as oxidative stress and mitochondrial oxidation in the liver of broiler chickens.

In experiment 1, DON-3S, OTA, and AFB_1_ were detected in the plasma of broiler chickens after the administration of a single oral bolus. The presence of these biomarkers in the plasma of broiler chickens, as well as their respective TK parameters, are in accordance with previous studies conducted on broiler chickens [25,27]. The appearance of a second peak of OTA in the plasma of broiler chickens at 8 h p.a. corresponds to previously described kinetics [25,27], and it is associated with the enterohepatic recirculation of OTA following its biliary excretion and reabsorption in the intestine [28]. The administration of the algoclay-based decontaminant along with the mycotoxins reduced the exposure to the three mycotoxins tested. The reduced exposure to AFB_1_ using a clay-based decontaminant could be expected, considering that bentonite clay is a registered additive for reducing the contamination by AFB_1_ in feed [29], and their modified forms have also been shown to effectively reduce exposure to AFB_1_ [30,31]. The reduction of exposure to OTA by a mycotoxin decontaminant in a TK model has not been demonstrated before. The adsorption of OTA on *Saccharomyces cerevisiae* extracts has been reported in vitro [32,33]. The present algoclay-based decontaminant also contains *S. cerevisiae* cell wall, which may be responsible for the decreased exposure to OTA in the present TK study. Absent or low effectiveness of DON adsorption has been reported, where the binding capacity ranged from 0 to 21% when exposing this mycotoxin to clay minerals or yeast cell-wall-derived products in aqueous solutions or bioassays [34], and only activated carbon could effectively bind 70–100% DON [34,35]. However, dietary supplementation with activated carbon is not indicated because it may bind nutrients [35] and cause intestinal damage [4]. Besides deactivation achieved by enzymes [35], an improved ability to bind DON was reported with modified clay minerals, reaching an adsorption ratio of 27.4% at intestinal pH [36]. In the present study, the algoclay-based decontaminant was able to decrease 40% of the oral bioavailability of DON. Multi-mycotoxin contamination is a common event in animal diets [37]. Hence, the capacity of a decontaminant to simultaneously reduce the exposure of the animals to several mycotoxins becomes relevant.

In experiment 2, the average body weight of birds at arrival was 40.0 g. Birds showed an average body weight higher than that expected in practice, being approximately 6, 137, and 80 g heavier than the expected weights in the Aviagen table for Ross broilers [38] for days 14 (541 g), 28 (1697 g), and 37 (2663 g), respectively. This indicates that the nutrients in the diet were within the required levels and that the birds were kept in optimal conditions for their growth. It was remarkable that birds fed the DON-contaminated diet, regardless of supplementation with the algoclay-based decontaminant, had a higher BWG on day 28 and a lower FCR on days 14 and 28 when compared to the control diet. When evaluating the nutrient contents of the diets, it was found that the contaminated corn batch had a higher crude protein level in its composition than the marginally contaminated one, resulting in contaminated diets with circa 1.5 g more of crude protein per kg than the marginally contaminated ones. Of the worldwide corn produced, 61% is directed for feed production, 13% for human consumption, and 16% for biofuel, whereas the use of the remaining 10% is variable [39]. One may expect that 26% of the corn not used for feed or food is probably rejected due to several factors, including high mycotoxin contamination. However, the food–feed competition, together with an expected decrease in crop corn yield of up to 29% in the upcoming decades due to climate change [40], should be taken into account when selecting batches for animal feeding. For instance, it is demonstrated herewith that the present *Fusarium* contamination did not impair the nutrient composition of the corn. Taking into account that climate change will also promote the proliferation of mycotoxins, field intervention to control *Fusarium* proliferation [41] and dietary supplementation with compounds that either directly or indirectly minimize the negative impact of mycotoxins [42,43,44] appear as a more sustainable solution.

Chronic exposure to DON led to a significant decrease in the VH in the jejunum of 37-day-old broiler chickens fed a DON-contaminated diet, despite no differences in production performance. At the tested DON concentrations and dietary conditions, it appears that prolonged exposure is required to affect intestinal villi cell renewal. It has been demonstrated that DON impairs enterocyte proliferation and slows down villi regeneration [3,4,45]. In the study from Wan et al. [45], the decrease in VH was accompanied by increased CD, probably because of an attempt by these progenitor cells to replace the enterocytes degenerating in the villi. In the present study, however, no increase in the CD was observed. This difference may be attributed to the type and time of exposure. In the present study, the birds were exposed to DON during the complete feeding period, i.e., 37 days; in the study from Wan et al. [45], birds were subjected to acute high exposure (10 mg/kg) for 7 days. The observed increase in CD during the initial week of exposure can likely be attributed to a compensatory mechanism. However, it is anticipated that CD would subsequently decrease after a prolonged period of exposure, as evidenced by previous studies [1,3,4,45,46], which reported a decline in CD after a minimum of 21 days of exposure to DON. It was noteworthy that villi damage occurred earlier on d28, focused on the villus tip (damage score 1). Villus contraction is a common occurrence following injury because the number of damaged cells exceeds the number of those that could occupy the empty positions, and this villus shortening consists of a mechanism to re-establish the epithelial barrier [47]. This explains the absence of villus damage on d37, but with villus shortening in the present study. When the contaminated diet was supplemented with the algoclay-based product, the damage score and VH were comparable to those observed in the jejunum of broiler chickens fed the control diet. It is important to bear in mind that the broiler chickens used in the present study were not challenged with diseases or other sources of intestinal harm. It is expected that a combination of dietary exposure to DON and other stress factors that affect gut health like subclinical coccidiosis, may increase intestinal harm [48].

In the performance trial, the birds had free access to feed, but reliable analysis of DON and its metabolites in the serum of the broiler chickens was not possible. For instance, the measured levels of DON were very low in all serum samples (<1.5 ng/mL), whereas in the TK study it was possible to recover DON-3S. These results highlight the difficulty of measuring biomarkers for exposure to DON in performance studies when feed intake is not controlled (time and quantity) and adds to the individual variability in absorption, distribution, metabolism, and excretion of the mycotoxin. The oral bolus model allows the administration of a controlled quantity of mycotoxin over time. Consequently, mycotoxins or their metabolites can be more easily detected and interpreted with better reliability. In the TK study, the algoclay-based decontaminant reduced the oral absorption of multiple mycotoxins, including DON, orally administered to broiler chickens in a single oral bolus model. Therefore, the observed effects on intestinal morphology in the present study could be explained by the adsorption of DON by the algoclay-based decontaminant. Furthermore, pelleting a diet can be a stressful process because the feed is exposed to temperatures ranging from 60 to 95 °C, which may affect the structure of some exogenous enzymes, resulting in their partial inactivation unless they are heat stable [49]. Enzymes can also be applied in liquid form after pelleting to avoid heat exposure, but this requires suitable application equipment in the feed mill. The positive results obtained in the present trial when the broiler chickens were fed pellet diets (experiment 2) suggest that the tested decontaminant is heat-stable during diet production, and does not require a system for application after pelleting.

Exposure to DON at a level below 3 mg/kg, combined with good management and diets with the required nutrient levels, should not result in liver metabolic disorders, but oxidative stress was not avoided. Of the evaluated markers for oxidative stress and liver metabolism, only GSS was upregulated (two-fold) on d37 when the broiler chickens were fed the DON-contaminated diet, and this effect was not observed when they were fed the DON-contaminated diet supplemented with the algoclay-based decontaminant. Glutathione is particularly concentrated in the liver, and its synthesis is dependent on enzymes, e.g., GSS. The expression of GSS is increased under stress conditions such as heat stress [11] or after exposure to DON [50]. The adsorption of DON on the algoclay-based decontaminant could explain the observed results, considering the algoclay-based decontaminant does not have a direct effect on liver parameters, since it is not orally absorbed but remains in the gut.

## 4. Conclusions

The administration of a single oral bolus containing 0.5 mg DON/kg BW, 0.25 mg OTA/kg BW, and 2.0 mg AFB_1_/kg BW to broiler chickens leads to systemic exposure to these mycotoxins. The concomitant administration of the algoclay-based decontaminant (2.5 g/kg feed) reduced systemic exposure to the three tested mycotoxins, by 39.9%, 44.3%, and 64.1%, respectively, for DON, OTA, and AFB_1_.

Dietary exposure to a corn-based diet with circa 3 mg/kg DON does not impair production performance in optimum housing conditions, but after 28 days of exposure it will cause villus damage in the jejunum, followed by shortening on day 37. Furthermore, oxidative stress takes place in the liver on day 37. Dietary supplementation with the algoclay-based decontaminant resulted in a positive impact on intestinal morphology and against oxidative stress. The present trial was performed with conventional broiler chickens. However, it is important to consider the potential risks associated with chronic exposure while raising slow-growing broiler chickens, since these chickens may be exposed to such risks through their feed for a period of 50–56 days.

## 5. Materials and Methods

Two independent experiments were performed in the present study. Both experiments involved an algoclay technology using the water-soluble polysaccharide ulvan, present in the cell wall of green seaweed (*Chlorophyceae*) of the genus *Ulva* and montmorillonite (layer clay). The algoclay technology consists of a blend of a water-soluble polysaccharide ulvan as the major compound, combined with bentonite clay and a yeast (*Saccharomyces cerevisiae*) cell wall to form the algoclay-based decontaminant. In experiment 1, a TK trial was performed to determine the ability of the algoclay-based decontaminant to detoxify DON, OTA, and AFB_1_. For this, plasma DON-3S, OTA, and AFB_1_ levels were measured. In experiment 2, a growth performance study was combined with blood and tissue sampling for the analysis of serum levels of DON and its main metabolite in poultry, DON-3S, as well as to evaluate intestinal morphology and the mRNA expression of markers related to oxidative stress and metabolism in the liver. Both studies were blindly performed at two different research institutes.

### 5.1. Experiment 1: Toxicokinetic Assay

#### 5.1.1. Chemicals, Products, and Reagents

The analytical standards of DON, OTA, and AFB_1_ were obtained from Fermentek (Jerusalem, Israel). The IS, ^13^C_15_-DON, ^13^C_20_-OTA, and ^13^C_17_-AFB_1_ were supplied by Biopure (Tulln, Austria). All standards were stored according to the recommendations of the supplier. Water, methanol (MeOH), ACN, glacial acetic acid, and formic acid for the mobile phases were of LC-MS grade and were obtained from Biosolve (Valkenswaard, The Netherlands).

#### 5.1.2. Broiler Chickens

To properly evaluate DON, DON-3S, OTA, and AFB_1_ circulating levels, a TK study was performed to determine the levels of these mycotoxins in the plasma of broiler chickens.

Sixteen healthy male broiler chickens (Ross 308) were included in this animal study. The chickens were obtained at 20 days old from a commercial breeder (Hatchery Vervaeke-Belavi, Tielt, Belgium). The study was conducted with the consent of the Ethical Committee of Poulpharm (EC number: 97_P22015-FP). The protocol used for this study is similar to that previously described by Devreese and colleagues [51]. Care, housing, and use of the animals were in compliance with Belgian (Belgian Royal Decree of 29 May 2013) and European (2010/63/EU) legislation on animal welfare and ethics.

The animals had one week of acclimatization; on day six of acclimatization, the animals were weighed. During the experimental phase of the study, the animals were housed per group of 8 birds in a floor pen of 2 m^2^. The group-housed chickens received mycotoxin control feed and water *ad libitum* throughout the complete experimental period. The feed used during the study was analyzed for the presence of mycotoxins through a multi-mycotoxin LC-MS/MS method (Primoris, Zwijnaarde, Belgium). The feed contained 188 µg/kg of DON and 2.6 µg/kg of AFB_1_ these levels are below the maximum guidance level constricted by EU regulations.

#### 5.1.3. Mycotoxin Administration and Blood Sampling

After a one-week acclimation period, 16 male, 27-days-old chickens (Ross 308) weighing approximately 1.6 kg BW were fasted for 12 h and then given a bolus containing either mycotoxins alone (*n* = 8) or mycotoxins and the algoclay-based decontaminant (*n* = 8) (0.5 mg DON/kg BW; 0.25 mg ochratoxin A/kg BW; 2.0 mg AFB_1_/kg BW; 2.5 g decontaminant/kg feed).

Blood samples of 1 mL were taken from the 16 chickens between 0 h (before bolus administration) and 24 h p.a. and were collected in heparinized tubes (Vacutest Kima, Novolab, Geraardsbergen, Belgium). Blood samples were centrifuged (3000× *g*, 10 min, 4 °C) and then the plasma was stored at −15 °C until analysis.

#### 5.1.4. Analysis Method

The plasma concentrations of DON, OTA, and AFB_1_ and the plasma response of DON-S were determined using LC-MS/MS, employing a validated method as described by [25,52].

For extraction, 150 µL of chicken plasma was aliquoted into the wells of an Oasis Ostro^®^ 96-well plate (Waters, Drinagh, Ireland), followed by spiking with 15 µL of a mixed internal standard (IS) solution (200 ng/mL of ^13^C_15_-DON, 100 ng/mL ^13^C_20_-OTA, and 20 ng/mL ^13^C_17_-AFB_1_). Subsequently, 450 µL of acetonitrile (ACN) with 0.1% formic acid was added; after gentle mixing, the Ostro^®^ plate was brought under vacuum (67.7 kPa) to eluate the analytes. The eluate was then dried under a gentle N_2_-stream at 40 ± 5 °C and reconstituted in 150 µL of methanol (MeOH)/water (85/15; *v*/*v*). An aliquot of 5 µL was injected into the Acquity H-Class UPLC system.

Chromatographic separation was achieved using an Acquity UPLC HSS T3 column (1.8 μm, 2.1 × 100 mm, Waters, Belgium) with a guard column of the same type (5 mm × 2.1 mm i.d., dp: 1.8 μm, Waters, Belgium). The mobile phases and gradient elution program were optimized for positive and negative electrospray ionization (ESI) modes. For ESI-positive mode, the mobile phases (MP) contained 0.3% formic acid in water (MP A) and 0.3% formic acid in MeOH (MP B). In the ESI-negative mode, the most suitable mobile phases consisted of 1% acetic acid in water (MP C) and 1% acetic acid in ACN (MP D). A gradient elution program was run for each ionization mode separately. For ESI-positive: 0–0.5 min (95% MP A, 5% MP B), 0.5–1.5 min (linear gradient to 60% MP B), 1.5–2.5 min (40% MP A, 60% MP B), 2.5–5.0 min (linear gradient to 80% MP B), 5.0–6.0 min (linear gradient to 99% MP B), 6.0–8.9 min (1% MP A, 99% MP B), 8.9–9.0 min (linear gradient to 95% MP A), and 9.0–12.0 min (95% MP A, 5% MP B). For ESI-negative: 0–1.5 min (95% MP C, 5% MP D), 1.5–3.0 min (linear gradient to 60% MP C), 3.0–4.0 min (60% MP C, 40% MP D), 4.0–7.0 min (linear gradient to 40% MP C), 7.0–9.0 min (40% MP C, 60% MP D), 9.0–9.5 min (linear gradient 95% MP C), and 9.5–12.0 min (95% MP C, 5% MP D). The flow rate was maintained at 300 μL/min, the column temperatures were set at 40 °C and the temperature of the autosampler was kept at 8 °C. The LC column effluent was analyzed using a Xevo tandem quadruple (TQ-S) mass spectrometer with specific instrument parameters set for desolvation gas flow: 800 L/h, desolvation temperature: 550 °C, cone gas flow: 150 L/h, source temperature: 150 °C, and capillary voltage: 3.0 kV for both ESI modes.

Quantification with MS was performed using specific transitions for each analyte. In the ESI-positive mode, DON (*m*/*z*) 297.00 > 249.10, ^13^C_15_-DON (*m*/*z*) 312.00 > 263.00, OTA (*m*/*z*) 404.00 > 238.90, ^13^C_20_-OTA (*m*/*z*) 424.00 > 250.00, AFB_1_ (*m*/*z*) 313.03 > 285.10, and ^13^C_17_-AFB_1_ (*m*/*z*) 330.10 > 255.10 transitions ions were used. For the ESI-negative mode, DON-S (*m*/*z*) 374.9 > 97.1 and ^13^C_15_-DON (*m*/*z*) 312.0 > 263.0 transitions ions were used. The limit of quantification (LOQ) for the components in this method was 0.5 ng/mL. The linearity of the components ranged from 0.50 to 100 ng/mL.

Only low concentrations of DON were quantifiable. Therefore, the TK parameters were calculated using the instrument response of the metabolite (DON-3S) and the concentrations of OTA and AFB_1_. The following parameters were calculated for each mycotoxin: area under the curve from time zero to the last point above the LOQ (AUC_0→t_), maximal plasma concentration or response (C_max_), and time at maximal plasma concentration or response (T_max_). The relative oral bioavailability ((average AUC_0→t_ mycotoxin + decontaminant/average AUC_0→t_ mycotoxin) × 100) was evaluated for each mycotoxin as a marker for the efficacy of the mycotoxin decontaminant. Furthermore, the effect of the mycotoxin decontaminant on the oral absorption of the mycotoxin was evaluated by comparing TK parameters between the mycotoxin and mycotoxin + decontaminant-treated chickens. A *t*-test was performed using SPSS 24.0 (IBM, Chicago, IL, USA) to evaluate possible significant differences in AUC_0→t_, C_max_ and T_max_.

### 5.2. Experiment 2: Naturally Contaminated Diet

#### 5.2.1. Broiler Chickens

A total of 600 one-day-old male Ross 308 broilers were divided into three dietary treatments of 200 chickens each (divided among 10 replicate pens with 20 chickens each). The birds were housed in 30 pens with wood shavings as the bedding material in the broiler facilities of Schothorst Feed Research (SFR), Lelystad, the Netherlands. Each pen (2.2 m^2^) had one feeder and three drinking nipples. Birds were kept until 37 days of age. Housing of the birds was performed according to EU legislation, and birds were monitored by a veterinarian throughout the experimental period. The birds were vaccinated against Newcastle Disease at d10 and against Infectious Bursal Disease at d20 of the trial.

#### 5.2.2. Diets and Experimental Design

The experiment comprised three dietary treatments. The control diet (Control) was prepared with a corn batch marginally contaminated with mycotoxins. Another basal diet was prepared with a corn batch naturally contaminated with DON (~6.8 mg/kg). This diet was split into two diets, i.e., a contaminated DON diet (DON) and a DON diet supplemented with the algoclay-based decontaminant, at a dosage of 2 g/kg diet (DON + algoclay-based decontaminant).

The recommended maximum level of DON in poultry diets is 5 mg/kg [2]. In the present study, the inclusion level of corn in the diet was 45% resulting in a final DON level between 2.69 and 2.91 mg/kg. The complete multi-mycotoxin analysis is given in Table 7. The mean levels of DON in the control diets during the starter, grower, and finisher phases were 0.19, 0.19, and 0.25 mg/kg, respectively. The mean levels of DON in the non-supplemented contaminated diets during the starter, grower, and finisher phases were 2.82, 2.69, and 2.71 mg/kg, respectively. The mean levels of DON in the contaminated diet supplemented with the algoclay-based decontaminant during the starter, grower, and finisher phases were 2.91, 2.66, and 2.81 mg/kg, respectively. Mycotoxin levels in the diets were determined by an independent and accredited (BELAC 057-TEST/ISO17025) laboratory (Primoris Holding, Ghent, Belgium) via liquid chromatography with tandem mass spectrometry (LC-MS/MS). In brief, a sample of each diet was subjected to a multi-mycotoxin analysis (24 mycotoxins). A description of the mycotoxins selected for analysis and the limit of quantification of each mycotoxin is given in the footnotes of Table 7.

All diets were prepared according to the nutritional requirements of broiler chickens (Table 8).

#### 5.2.3. Growth Performance

Broilers were weighed per pen on d0, d14, d28, and d37. Feed consumption and mortality were recorded throughout the experimental period. Body weight gain (BWG), feed intake (FI) and feed conversion ratio (FCR) were determined in the cumulative phases from d0–14, d14–28, d28–37, and the overall period of d0–37.

#### 5.2.4. Jejunum Morphometry and Scoring

Samples of jejunum from a bird per pen (10 chickens per treatment) at d14, 28, and 37 were collected and fixed in buffered formalin for histological analysis. The prepared histological sections were stained with periodic acid–Schiff (PAS) counterstaining with haematoxylin staining and scanned using the NanoZoomer-XR (Hamamatsu Photonics KK, Hamamatsu, Japan). The scanned slides were viewed through the viewer software (NDP.view2; Hamamatsu, Version 2.27.25) and analyzed using the analysis software (NDP.analyze; Hamamatsu). The illus height (VH), crypt depth (CD), and villus area (µm^2^) of each individual bird were measured (15 villi per intestinal segment). The measurements of VH and CD were used to calculate the VH:CD ratio. Only the intact villi were measured. Measurements were double-blinded and performed by a trained veterinarian.

To evaluate the degree of mucosal damage, the mucosa was classified as degree 0 if presenting an intact structure with no visible damage to degree 6 if severely damaged, as previously described [53].

#### 5.2.5. mRNA Expression of Markers for Liver Function

A liver sample was collected from the same sampled birds for jejunum collection (10 chickens per treatment) and submitted to RNA isolation using the SV Total RNA Isolation System (Promega, Madison, WI, USA) and subsequent cDNA Synthesis (Bio-Rad, Hercules, CA, USA) according to the manufacturer’s instructions. Primers, as presented in Table 9, were commercially produced (Eurogentec, Maastricht, The Netherlands). qPCR was performed using the MyIQ single-color, real-time PCR detection system (Bio-Rad) and MyiQ System Software Version 1.0.410 (Bio Rad Laboratories Inc., Hercules, CA, USA). Data were analyzed using the efficiency-corrected DeltaDelta-Ct method [54]. The fold-change values of the genes of interest were normalized using the geometric mean of the fold-change values of hypoxanthine-guanine phosphoribosyl transferase (HPRT) and β-actin (ACTB). The mRNA expression of markers in the liver was selected based on their role, i.e., oxidative stress (GSS and iNOS) and metabolism (CPT1 and HMGCR).

#### 5.2.6. Serum and Plasma Analysis of DON and DON-3S

After euthanasia for tissue sampling at days 14, 28, and 37 (a bird per pen; 10 chickens per treatment), blood samples (4 mL) were collected from the broiler chickens. The blood sampling took place every morning for two hours after the lights were turned on to stimulate feed intake. Serum was harvested via the centrifugation of blood at 1500× *g* for 10 min at 4 °C and sent for the analysis of DON and DON-3 sulphate levels [26].

### 5.3. Statistical Analysis

The pen was the experimental unit for all data. The experimental data were analyzed using ANOVA (GenStat Version 22.0, 2022, Hemel Hempstead, UK). Treatment means were compared with Tukey’s post hoc test. Values with *p* ≤ 0.05 were considered statistically significant. The *p*-value and SEM (standard error of the mean) are given per response parameter.

## Figures and Tables

**Figure 1 toxins-16-00207-f001:**
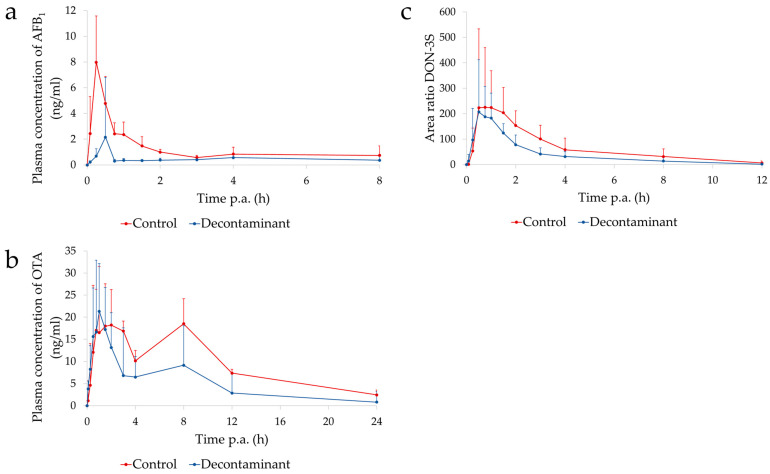
Mean (±SD) plasma concentration–time curve of AFB_1_ (**a**) and OTA (**b**) and instrument response–time curve of DON-3S (**c**) up to 8 h (AFB_1_), 24 h (OTA), and 12 h (DON-3S) after an oral bolus administration of AFB_1_ (2.0 mg/kg BW), OTA (0.25 mg/kg BW), and DON (0.5 mg/kg BW) with and without the algoclay-based decontaminant; p.a. = post-administration; red = control, and blue = decontaminant.

**Figure 2 toxins-16-00207-f002:**
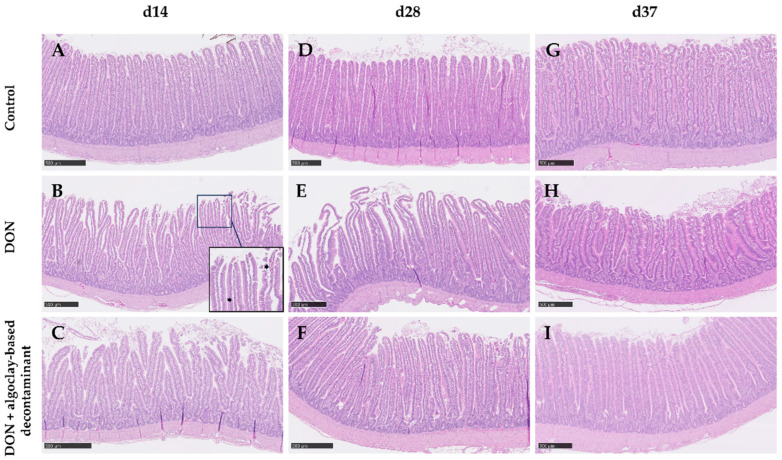
Illustrative images of PAS-haematoxylin-stained sections of jejunum from broiler chickens fed the experimental diets. (**A**–**C**): Jejunum sections of 14-day-old broiler chickens fed the control, DON, and DON + algoclay-based decontaminant diets, respectively. In (**B**), an insert shows haemorrhagic areas in the villus (black arrows). (**D**–**F**): Jejunum sections of 28-day-old broiler chickens fed the control, DON, and DON + algoclay-based decontaminant diets, respectively. In Panels (**D**,**E**), it is possible to observe damage in the villus tip of the jejunum. (**G**–**I**): Jejunum sections of 37-day-old broiler chickens fed the control, DON, and DON + algoclay-based decontaminant diets, respectively. Scale bars = 500 μm.

**Table 1 toxins-16-00207-t001:** Toxicokinetic characteristics of AFB_1_ after a single oral bolus administration of DON, OTA, and AFB_1_ whether combined or not with the algoclay-based decontaminant.

Toxicokinetic Parameter	Mycotoxins	Mycotoxins + Algoclay-Based Decontaminant	*p*-Value	SEM
AUC_0–8h_ (h.ng/mL)	9.44 ^b^	3.39 ^a^	<0.001	0.98
C_max_ (ng/mL)	8.65 ^b^	2.07 ^a^	<0.01	1.72
T_max_ (h)	0.26 ^b^	3.44 ^a^	0.03	1.21
Relative F (%)	100	35.9		

^a,b^ Values without a common letter within a row differ significantly (*p* < 0.05). SEM: standard error of the mean; AUC: area under the curve; C_max_: maximal plasma concentration; T_max_: time at maximal plasma concentration; Relative F: relative oral bioavailability. The experiment was carried out with eight repetitions, where each repetition represents one broiler chicken.

**Table 2 toxins-16-00207-t002:** Toxicokinetic characteristics of OTA after a single oral bolus administration of DON, OTA, and AFB_1_ whether or not combined with the algoclay-based decontaminant.

Toxicokinetic Parameter	Mycotoxins	Mycotoxins + Algoclay-Based Decontaminant	*p*-Value	SEM
AUC_0–24h_ (h.ng/mL)	205.99 ^b^	114.74 ^a^	<0.001	18.48
C_max_ (ng/mL)	31.45	26.48	0.37	5.38
T_max_ (h)	3.28	1.75	0.31	1.45
Relative F (%)	100	55.7		

^a,b^ Values without a common letter within a row differ significantly (*p* < 0.05). SEM: standard error of the mean; AUC: area under the curve; C_max_: maximal plasma concentration; T_max_: time at maximal plasma concentration; Relative F: relative oral bioavailability. The experiment was carried out with eight repetitions, where each repetition represents one broiler chicken.

**Table 3 toxins-16-00207-t003:** Toxicokinetic characteristics of DON-3 sulphate after a single oral bolus administration of DON, OTA, and AFB_1_ whether combined or not with the algoclay-based decontaminant.

Toxicokinetic Parameter	Mycotoxins	Mycotoxins + Algoclay-Based Decontaminant	*p*-Value	SEM
AUC_0–12h_ (response.h)	760.9 ^b^	457.6 ^a^	0.01	92.9
C_max_ (response)	346.1	270.1	0.48	105.0
T_max_ (h)	1.31	0.91	0.25	0.33
Relative F (%)	100	60.1		

^a,b^ Values without a common letter within a row differ significantly (*p* < 0.05). SEM: standard error of the mean; AUC: area under the curve; C_max_: maximal plasma response; Relative F: relative oral bioavailability; T_max_: time at maximal plasma response. The experiment was carried out with eight repetitions, where each repetition represents one broiler chicken.

**Table 4 toxins-16-00207-t004:** Effect of the dietary treatments on broiler performance and mortality rate.

	Diets			*p*-Value	SEM
	Control	DON Diet	DON + Algoclay-Based Decontaminant		
d0–14					
BW d14	534	550	557	0.06	6.4
BWG (g)	494	510	517	0.06	6.38
FI (g)	553	549	554	0.89	7.33
FCR (g/g)	1.119 ^b^	1.076 ^a^	1.071 ^a^	<0.0001	0.0064
Mortality (%)	1.0	2.0	0.5	0.49	0.88
d14–28					
BW d28	1774 ^a^	1859 ^b^	1870 ^b^	0.01	19.9
BWG (g)	1240 ^a^	1309 ^b^	1312 ^b^	0.01	15.6
FI (g)	1692	1753	1749	0.08	19.3
FCR (g/g)	1.366 ^b^	1.340 ^a^	1.332 ^a^	0.01	0.0070
Mortality (%)	0.5	1.0	1.5	0.58	0.66
d28–37					
BW d37	2694	2762	2773	0.12	27.6
BWG (g)	920	903	903	0.78	19.6
FI (g)	1566	1599	1575	0.56	22.0
FCR (g/g)	1.705	1.775	1.749	0.14	0.0234
Mortality (%)	0.5	1.5	1.5	0.67	0.88
d0–37					
BWG (g)	2654	2722	2733	0.12	27.6
FI (g)	3811	3901	3877	0.23	36.9
FCR (g/g)	1.436	1.433	1.419	0.16	0.0062
Mortality (%)	2.0	4.5	3.5	0.49	1.45

^a,b^ Values without a common letter within a row differ significantly (*p* < 0.05). SEM: standard error of the mean; BW: body weight; BWG: body weight gain; FI: feed intake; FCR: feed conversion ratio. The experiment was carried out with 10 repetitions, where each repetition consisted of a pen with 20 chickens.

**Table 5 toxins-16-00207-t005:** Effect of the treatments on morphometric parameters and morphologic scores of jejunum.

	Diets			*p*-Value	SEM
	Control	DON Diet	DON + Algoclay-Based Decontaminant		
d14					
Villus height (µm)	730	613	703	0.19	45.2
Crypt depth (µm)	182	178	192	0.62	10.2
VH:CD	4.17	3.58	3.77	0.16	0.21
Villus area (mm^2^)	96	77	78	0.30	9.21
Score	0.54	0.83	0.84	0.13	0.11
d28					
Villus height (µm)	990	765	873	0.15	76.4
Crypt depth (µm)	200	189	224	0.19	13.2
VH:CD	5.06	4.27	4.18	0.13	0.32
Villus area (mm^2^)	135	116	152	0.26	15.1
Score	0.51 ^a^	1.14 ^b^	0.68 ^a,b^	0.04	0.16
d37					
Villus height (µm)	1187 ^b^	884 ^a^	1030 ^a,b^	0.01	59.3
Crypt depth (µm)	224	225	201	0.50	15.9
VH:CD	5.58 ^b^	4.13 ^a^	5.41 ^b^	0.02	0.33
Villus area (mm^2^)	96	77	78	0.67	24.4
Score	0.69	0.93	0.53	0.24	0.16

^a,b^ Values without a common letter within a row differ significantly (*p* < 0.05). SEM: standard error of the mean; VH:CD: villus height: crypt depth ratio. The experiment was carried out with 10 repetitions, where each repetition consisted of a sampled chicken per experimental pen. A total of 15 villi were evaluated per intestinal segment.

**Table 6 toxins-16-00207-t006:** Mean (±SEM) fold-change expression of glutathione synthetase (GSS), nitric oxide synthase (iNOS), and carnitine palmitoyltransferase-1 (CPT1) in the liver of broiler chickens fed the experimental diets.

	Diets			*p*-Value	SEM
	Control	DON Diet	DON + Algoclay-Based Decontaminant		
d14					
GSS	1.00	0.84	0.93	0.61	0.12
iNOS	1.00	1.06	0.95	0.66	0.09
CPT1	1.00	0.99	1.10	0.74	0.11
d28					
GSS	1.00	0.69	0.57	0.68	0.35
iNOS	1.00	1.03	1.16	0.81	0.17
CPT1	1.00	0.68	1.29	0.11	0.19
d37					
GSS	1.00 ^a^	2.01 ^b^	1.05 ^a^	0.01	0.20
iNOS	1.00	1.09	0.89	0.27	0.09
CPT1	1.00	1.32	1.20	0.79	0.32

^a,b^ Values without a common letter within a row differ significantly (*p* < 0.05). SEM: standard error of the mean; CPT1: carnitine palmitoyltransferase-1; GSS: glutathione synthetase; iNOS: nitric oxide synthase. The experiment was carried out with 10 repetitions, where each repetition consisted of a sampled chicken per experimental pen.

**Table 7 toxins-16-00207-t007:** Levels (mg/kg) of mycotoxins in the experimental diets during the different feeding periods.

Mycotoxins (mg/kg)	Control	DON	DON + Algoclay-Based Decontaminant
		Starter (d0–14)	
DON	0.19	2.82	2.91
3 + 15 Ac-DON	<LOQ	0.06	0.10
DON-3G	0.06	0.40	0.51
Zearalenone	<LOQ	0.16	0.20
Fumonisins B1 + B2	0.05	0.29	0.29
Beauvericin	0.01	0.02	0.02
Enniatin B	0.02	0.02	0.02
Enniatin B1	0.01	0.01	0.01
		Grower (d14–28)	
DON	0.19	2.69	2.66
3 + 15 Ac-DON	<LOQ	0.08	0.11
DON-3G	0.04	0.37	0.44
Zearalenone	<LOQ	0.26	0.23
Fumonisins B1 + B2	<LOQ	0.19	0.09
Beauvericin	0.01	0.02	0.02
Enniatin B	0.02	0.02	0.02
Enniatin B1	0.01	0.01	0.01
		Finisher (d28–37)	
DON	0.25	2.71	2.81
3 + 15 Ac-DON	<LOQ	-	0.11
DON-3G	0.06	0.49	0.49
Zearalenone	<LOQ	0.23	0.24
Fumonisins B1 + B2	0.04	0.08	0.13
Beauvericin	0.01	0.02	0.02
Enniatin B	0.03	0.03	0.03
Enniatin B1	0.01	0.01	0.01

Analyzed mycotoxins with their respective limit of quantification (LOQ): Aflatoxin B_1_ (1 µg/kg), Aflatoxin B2 (1 µg/kg), Aflatoxin G1 (1 µg/kg), Aflatoxin G2 (1 µg/kg), Alternariol (2 µg/kg), Alternariol monomethyl ether (2 µg/kg), Beauvericin (5 µg/kg), Citrinin (10 µg/kg), Cytochalasine E (2 µg/kg), Deoxynivalenol (20 µg/kg), 3 + 15 Ac-deoxynivalenol (20 µg/kg), Deoxynivalenol-3-glucoside (20 µg/kg), Diacetoxyscirpenol (5 µg/kg), Enniatin A (5 µg/kg), Enniatin A1 (5 µg/kg), Enniatin B (5 µg/kg), Enniatin B1 (5 µg/kg), Fumonisins B1 + B2 (20 µg/kg), Moniliformin (5 µg/kg), Nivalenol (50 µg/kg), Ochratoxin A (1 µg/kg), Roquefortine C (5 µg/kg), Sterigmatocystin (1 µg/kg), T-2/HT-2 toxin (10 µg/kg), and Zearalenone (15 µg/kg).

**Table 8 toxins-16-00207-t008:** Dietary composition and calculated nutrients.

Ingredients (%)	Starter (d0–14)	Grower (d14–28)	Finisher (d28–37)
Corn	45.00	45.00	45.00
Soybean meal	33.55	29.55	24.81
Wheat	9.71	12.50	16.55
Barley	5.00	5.00	5.00
Soybean oil	0.00	0.08	0.34
Animal fat	3.23	4.36	4.79
Salt	0.36	0.23	0.11
Limestone	0.80	0.80	0.82
Monocalcium Phosphate	1.13	1.15	1.19
Sodium Bicarbonate	0.00	0.19	0.37
Lysine HCl	0.26	0.24	0.21
DL-Methionine	0.31	0.26	0.20
Threonine	0.08	0.06	0.06
Tryptophane	0.002	0.00	0.00
Valine	0.03	0.01	0.01
Choline chloride	0.06	0.06	0.06
Vitamin and mineral premix	0.50	0.50	0.50
Nutrients			
AMEn, kcal/kg	2.900	3.000	3.075
DM, g/kg	880	881	882
Ash, g/kg	52.51	49.94	47.37
Crude protein, g/kg	218	201	182
Crude fat, g/kg	57.90	69.99	77.01
Crude fiber, g/kg	22.81	22.11	21.44
Starch, g/kg	368	385	408
Sugar, g/kg	31.18	29.33	27.34
Ca, g/kg	6.00	5.95	5.95
P, g/kg	6.12	6.01	5.93
Mg, g/kg	1.64	1.56	1.47
K, g/kg	9.51	8.74	7.87
Na, g/kg	1.50	1.50	1.50
Cl, g/kg	3.26	2.46	1.66
dEB, meq	217	220	220

AMEn: Nitrogen-corrected apparent metabolizable energy; Ca: Calcium; Cl: Chlorine; dEB: Dietary electrolyte balance; DM: Dry matter; K: Potassium; Mg: Magnesium; Na: Sodium; P: Phosphorus.

**Table 9 toxins-16-00207-t009:** Primers used for the quantification of genes of interest (GOI) and housekeeping gene (HKG) expression in the liver of broiler chickens.

Genes	Primer Sequence	Annealing Tº	Reference
HKG			
ACTB	F: ATGTGGATCAGCAAGCAGGAGTA R: TTTATGCGCATTTATGGGTTTTGT	61	[55]
HPRT	F: CGTTGCTGTCTCTACTTAAGCAG R: GATATCCCACACTTCGAGGAG T	61	[6]
GOI			
Oxidative stress			
GSS	F: GTGCCAGTTCCAGTTTTCTTATG R: TCCCACAGTAAAGCCAAGAG	61.0	[10]
iNOS	F: GGACAAGGGCCATTGCACCA R: TCCATCAGCGCTGCGCACAA	61.0	[56]
Metabolism			
CPT1	F: AAGGGTACAGCAAAGAAGATCCA R: CCACAGGTGTCCAACAATAGGAG	61.0	[57]
HMGCR	F: TTGGATAGAGGGAAGAGGGAAG R: CTCGTAGTTGTATTCGGTAA	55.7	[58]

CPT1: carnitine palmitoyltransferase-1; GSS: glutathione synthetase; HMGCR: 3-hydroxy-3-methylglutaryl-coenzyme A reductase; iNOS: nitric oxide synthase.

## Data Availability

The data presented in this study are available in this article.

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
