# Peer review of "An Algoclay-Based Decontaminant Decreases Exposure to Aflatoxin B1, Ochratoxin A, and Deoxynivalenol in a Toxicokinetic Model, as well as Supports Intestinal Morphology, and Decreases Liver Oxidative Stress in Broiler Chickens Fed a Diet Naturally Contaminated with Deoxynivalenol"

_toxins, 2024, doi:10.3390/toxins16050207_

Round 1

Reviewer 1 Report

Comments and Suggestions for Authors

The manuscript determined the effect of algoclay on the performance, intestinal integrity, liver oxidative stress and metabolism, and mycotoxin decontamination in broiler chickens fed a DON-contaminated diet. The study designed well and obtained some interesting information. The following are my comments, which need to be addressed before it can be accepted.

Line 111, what is the meaning of the number “6,137”? In addition, it is suitable to move the paragraph (lines 110-115) to the “Discussion” section.

In the Table 1, please provide the meanings represented by the letters a and b in the note of the table, as well as the differences in meanings between the upper and lower case letter a.

As elucidated in line 129, no significant differences in villus morphometry of jejunum were observed on days 14 and 28, on d37. However, in Figure 1, it can be clearly seen that there are significant changes in the morphology of the jejunum on day 14 and day 28 compared to the control group.

Line 153, “14-day-old” should be “37-day-old”.

What is the number of samples determined in which the data in Table 2 and Table 3 were from? Provide the number in the section of “Results” or “Materials and Methods”

Line 176, add a space between the words “DON-3Scould”

Figure 2 is not mentioned in the text description, add it please.

In the section of Discussion (Lines 216-217), the means of dietary supplementation with compounds to minimize the negative impact of mycotoxins was briefly introduced. To arouse readers' interest, it is necessary to clearly introduce some substances with detoxification effects, such as curcumin, yeast cell wall extract, and allow me to suggest the following publication to be cited herein, please.

Zhang, J.; Sun, X.; Chai, X.; Jiao, Y.; Sun, J.; Wang, S.; Yu, H.; Feng, X. Curcumin mitigates oxidative damage in broiler liver and ileum caused by AFB1-contaminated feed through Nrf2 signaling pathway. Animals. 2024, 14, 409. https://doi.org/10.3390/ani14030409.

Weaver, A.C.; Weaver, D.M.; Adams, N.; Yiannikouris, A. Use of yeast cell wall extract for growing pigs consuming feed contaminated with mycotoxins below or above regulatory guidelines: a meta-analysis with meta-regression. Toxins 2023, 15, 596. https://doi.org/10.3390/toxins15100596.

Lines 280-282, “Furthermore, a combination of dietary exposure to DON and other stress factors that affect gut health like subclinical coccidiosis, may increase the intestinal harm”, the sentence is not appropriate in the Conclusion section because this study does not cover the topic of subclinical  coccidiosis. Therefore, I suggest deleting the sentence or moving it to the Discussion section to expand on the topic.

Supplementary Table S1 was missing, provide it please.

As presented in lines 356-357, “the blood sampling took place every morning two hours after the lights were on to stimulate feed intake”. However, generally, experimental animals are fasted for about 12 hours before tissue samples are collected to reduce the impact of feeding on experimental data, such as body weight and blood biochemical indicators. In this experiment, why the birds were stimulated to feed before sampling?

Line 364, what does the meaning of the abbreviation “WW”?

In the toxicokinetic study (lines 360-365), the oral bolus model was established using AFB1, DON ant OTA. I am puzzled why the establishment of animal models requires the addition of two other mycotoxins while this study is aimed at studying the toxicokinetics of DON. Could the authors give the explanation?

Comments on the Quality of English Language

Minor editing of English language required.

Author Response

Reviewer #1 

The manuscript determined the effect of algoclay on the performance, intestinal integrity, liver oxidative stress and metabolism, and mycotoxin decontamination in broiler chickens fed a DON-contaminated diet. The study designed well and obtained some interesting information.

A: We would like to acknowledge the present reviewer for the pertinent questions and remarks. To make clear that this present study consists of two independent experiments, the Materials and Methods and Results sections were rewritten.

The following are my comments, which need to be addressed before it can be accepted.

Line 111, what is the meaning of the number “6,137”? In addition, it is suitable to move the paragraph (lines 110-115) to the “Discussion” section.

A: 6, 137, and 80 g heavier than the expected weight for days 14, 28, and 37, respectively. This paragraph is now in the discussion section (Lines 288-289).

In the Table 1, please provide the meanings represented by the letters a and b in the note of the table, as well as the differences in meanings between the upper and lower case letter a.

A: Please, accept our apologies for the missing information. Footnotes from all tables were checked and corrected/included when needed.

As elucidated in line 129, no significant differences in villus morphometry of jejunum were observed on days 14 and 28, on d37. However, in Figure 1, it can be clearly seen that there are significant changes in the morphology of the jejunum on day 14 and day 28 compared to the control group.

A: Indeed. As informed in the Materials and Methods section, only intact villi were measured. Therefore, these villi with morphological alterations were not considered when measuring their length. Damage in villi was increased (as observed by a significant increase in damage score on day 28), as observed in Figure 1. On day 37, the damage score difference is not observed anymore, but the villus height is significantly decreased. As pointed out in the Discussion section, “It was noteworthy that villi damage occurred earlier, on d28, being focused on the villus tip (damage score 1). Villus contraction is a common occurrence following injury because the number of damaged cells exceeds the number of those that could occupy the empty positions, and this villus shortening consists of a mechanism to re-establish the epithelial barrier [47]. This explains the absence of villus damage on d37 but with a villus shortening in the present study. (Lines 327-332)”

Line 153, “14-day-old” should be “37-day-old”.

A: It is now corrected (Line 228).

What is the number of samples determined in which the data in Table 2 and Table 3 were from? Provide the number in the section of “Results” or “Materials and Methods”

A: This information is now given in the Material and Methods section, as well as in the footnote of the tables.

Line 176, add a space between the words “DON-3Scould”

A: Done (Line 252).

Figure 2 is not mentioned in the text description, add it please.

A: This is now Figure 1 and in the current manuscript Figure 1 is referred to in Lines 136, 155 and 166.

In the section of Discussion (Lines 216-217), the means of dietary supplementation with compounds to minimize the negative impact of mycotoxins was briefly introduced. To arouse readers' interest, it is necessary to clearly introduce some substances with detoxification effects, such as curcumin, yeast cell wall extract, and allow me to suggest the following publication to be cited herein, please.

Zhang, J.; Sun, X.; Chai, X.; Jiao, Y.; Sun, J.; Wang, S.; Yu, H.; Feng, X. Curcumin mitigates oxidative damage in broiler liver and ileum caused by AFB1-contaminated feed through Nrf2 signaling pathway. Animals. 2024, 14, 409. https://doi.org/10.3390/ani14030409.

Weaver, A.C.; Weaver, D.M.; Adams, N.; Yiannikouris, A. Use of yeast cell wall extract for growing pigs consuming feed contaminated with mycotoxins below or above regulatory guidelines: a meta-analysis with meta-regression. Toxins 2023, 15, 596. https://doi.org/10.3390/toxins15100596.

A: We acknowledge the suggestions and these references are now added to the manuscript (References 43 and 44).

Lines 280-282, “Furthermore, a combination of dietary exposure to DON and other stress factors that affect gut health like subclinical coccidiosis, may increase the intestinal harm”, the sentence is not appropriate in the Conclusion section because this study does not cover the topic of subclinical  coccidiosis. Therefore, I suggest deleting the sentence or moving it to the Discussion section to expand on the topic.

A: This sentence is now in the Discussion section (Lines 334-338).

Supplementary Table S1 was missing, provide it please.

A: This Table was given in a supplementary file. It is now found in the body of the manuscript (Table 8).

As presented in lines 356-357, “the blood sampling took place every morning two hours after the lights were on to stimulate feed intake”. However, generally, experimental animals are fasted for about 12 hours before tissue samples are collected to reduce the impact of feeding on experimental data, such as body weight and blood biochemical indicators. In this experiment, why the birds were stimulated to feed before sampling?

A: Fasting animals is a procedure indicated for some types of trials and types of blood biochemistry analysis. In experiment 2, the broiler chickens were fed ad libitum, and mycotoxin administration did not occur via a highly concentrated bolus but via natural feed intake. Furthermore, intestinal tissue was collected for morphological analysis. Fasting chickens for more than 8 hours may affect intestinal morphology and could result in a misleading factor when performing damage scoring. In experiment 1, a toxicokinetic experiment was performed, where a bolus of contaminated material was administered to the broiler chickens. In this case, a 12-hour fast before starting the trial is indicated.

Line 364, what does the meaning of the abbreviation “WW”?

A: Our apologies. Instead of WW, is BW. (Line 422).

In the toxicokinetic study (lines 360-365), the oral bolus model was established using AFB1, DON ant OTA. I am puzzled why the establishment of animal models requires the addition of two other mycotoxins while this study is aimed at studying the toxicokinetics of DON. Could the authors give the explanation?

A: This study aimed to evaluate the impact of the algoclay-based decontaminant on all three mycotoxins. As these mycotoxins undergo passive diffusion through absorption processes, no interactions are expected among them, enabling their combination in a single trial. Consequently, this approach reduces the number of experiments required from three (one for each toxin separately) to just one due to the availability of a naturally contaminated feedstuff.

Reviewer 2 Report

Comments and Suggestions for Authors

The paper describes the results of an experiment on a DON-contaminated diet of broiler chickens in terms of performance, intestinal integrity, liver oxidative stress and metabolism. The paper is written well. The abstract gives a clear overview of the contents of the paper, and all sections give detailed information on the different parts of the analysis, in a way that is still pleasant to read. I have only some minor comments:

- please start with the full names of FI, BWG and FCR in lines 116-117, and then use the shortnames afterwards

- I guess the results of table 4 relate to figure 2, but I missed the details (e.g. what is Time p.a., how does the Cmax relates to the Area ratio DON-3S?)

- the term 'chicks' must be replaced by the term 'chickens'

I send these comments also to the Editors

Author Response

Reviewer #2

The paper describes the results of an experiment on a DON-contaminated diet of broiler chickens in terms of performance, intestinal integrity, liver oxidative stress and metabolism. The paper is written well. The abstract gives a clear overview of the contents of the paper, and all sections give detailed information on the different parts of the analysis, in a way that is still pleasant to read. I have only some minor comments:

A: We would like to acknowledge the present reviewer for the suggestions and comments.

Please start with the full names of FI, BWG and FCR in lines 116-117, and then use the shortnames afterwards

A: All the full names are now given before abbreviations.

- I guess the results of table 4 relate to figure 2, but I missed the details (e.g. what is Time p.a., how does the Cmax relates to the Area ratio DON-3S?)   
A: All this information is now given in the Material and Methods section. Time p.a. stands for time post-administration, while Cmax represents the maximum plasma concentration or response (as indicated by its unit). The Cmax of the Area ratio of Don-3S signifies the anticipated highest mean response (n=8) of the mycotoxin in the plasma of the chickens, derived through non-compartmental analysis. The Area ratio is alternatively referred to as the response (unitless).

- the term 'chicks' must be replaced by the term 'chickens'

A: It is now corrected.

Reviewer 3 Report

Comments and Suggestions for Authors

The problem of the presence of mycotoxins in feed is very important for the health of animals and their productivity. It is also very important to assess the effectiveness of detoxifiers and the possibility of their use in the process of feeding animals with feed contaminated with mycotoxins. There are few studies on live animals, and these provide the best and most reliable results. Therefore, the topic should be assessed as very interesting and important for the health and productivity of animals.

However, the way the experiment was conducted and described raises serious doubts. In principle, it can be concluded that only the histopathological evaluation was performed and described correctly.

Comments on the experiment methodology:

Naturally contaminated corn was used and the mycotoxin concentrations that were described do not appear to be natural contamination. Very high concentration of DON and low ZEN and no (below the detection level of T-2 toxin, HT-2, ochratoxin and nivalenol actually do not occur. Please explain.

The 45% corn content in the feed results in a very high energy level in the feed - which may cause significant changes in liver metabolism. Please describe the nutritional value of the feed and explain it.

The high concentrations of DON used in the experiment make it possible to determine DON in broiler chicken serum. This is the most reliable documentation of the effectiveness of biogegradation or sorption of a toxin. The determination of DON in broiler chicken serum occurs already at the level of 1 mg DON/kg feed.

The entire chapter describing a parallel experiment examining the toxicokinetics of mycotoxins administered in a bolus form is downright grotesque and unbelievable. It lacks reliable methodological information and a description of the results in accordance with the standards. Please contact a toxicologist to prepare a reliable and true description in this chapter.

Comments to the editor of the text:

The title contains information about an intestinal integrity test - no such test took place. The structure of the intestinal villi was assessed and properly described. And this is the only value of this manuscript. All elements of toxicological tests are described incorrectly, and the results are unreliable.

DON is very little susceptible to sorption. Most feed additives used in nutrition are based on the biodegradation of DON by enzymes. No explanation as to why sorption should work so effectively in this experiment. Please explain.

Author Response

Reviewer #3

The problem of the presence of mycotoxins in feed is very important for the health of animals and their productivity. It is also very important to assess the effectiveness of detoxifiers and the possibility of their use in the process of feeding animals with feed contaminated with mycotoxins. There are few studies on live animals, and these provide the best and most reliable results. Therefore, the topic should be assessed as very interesting and important for the health and productivity of animals. However, the way the experiment was conducted and described raises serious doubts. In principle, it can be concluded that only the histopathological evaluation was performed and described correctly. Naturally contaminated corn was used and the mycotoxin concentrations that were described do not appear to be natural contamination. Very high concentration of DON and low ZEN and no (below the detection level of T-2 toxin, HT-2, ochratoxin and nivalenol actually do not occur. Please explain.

A: The present study consists of two independent experiments. In experiment 1, a toxicokinetic study was performed. In the present version, we give all the results obtained with the three mycotoxins present in the bolus. In experiment 2 (growth performance, intestinal morphology, and liver mRNA expression), we used a naturally contaminated diet. To clarify this, the Materials and Methods section was rewritten and the two experiments were separately described. It is not ethical to forge data or the source of contamination in a diet, and we have no reason for such an appalling act. After more than 10 years of analysing feedstuffs contaminated with mycotoxins, it is very common to detect some feedstuffs with high levels of DON and low (or even absent) ZEN levels in corn, wheat, or other feedstuffs. For example, soy hulls can be highly contaminated with ZEN without the presence of DON, and sugar beet pulp can be highly contaminated with ZEN with low levels of DON. To help the referee, we are attaching some of our intern data from the analysis of several feedstuffs. These are analyses performed for several feedstuffs that were not used in the present trial and are presented as examples. Additionally, we add also all the analyses of the diets used in the present trial.

The 45% corn content in the feed results in a very high energy level in the feed - which may cause significant changes in liver metabolism. Please describe the nutritional value of the feed and explain it.

A: When a diet is properly formulated, 45% corn in the feed will not result in a high energy level leading to changes in liver metabolism, especially in broiler chickens. If the diet is not well balanced and exceeds in energy, then fatty liver syndrome will be observed in laying hens. The Table with the dietary composition and nutrient levels, including energy, was previously given as a Supplementary Table, but now it is given in the body of the manuscript (Table 8).

The high concentrations of DON used in the experiment make it possible to determine DON in broiler chicken serum. This is the most reliable documentation of the effectiveness of biogegradation or sorption of a toxin. The determination of DON in broiler chicken serum occurs already at the level of 1 mg DON/kg feed. 

A: To effectively evaluate the impact of the binder, it's imperative to achieve an adequately high plasma concentration. If your control group's plasma concentration hovers just above the Limit of Quantitation (LOQ), it becomes challenging to discern any reduction in absorption facilitated by the binder. Additionally, the dosage of DON administered as a bolus in Experiment 1 aligns with the European Union's feed limit of 5 ppm.

The entire chapter describing a parallel experiment examining the toxicokinetics of mycotoxins administered in a bolus form is downright grotesque and unbelievable. It lacks reliable methodological information and a description of the results in accordance with the standards. Please contact a toxicologist to prepare a reliable and true description in this chapter.

A: Indeed, the description of the toxiconetics study was not clear, and all details are now given. Furthermore, the complete Materials and Methods section was rewritten to properly describe the two independent experiments. We understand that the referee misses reliable methodological information. We appreciate criticisms, but it is not acceptable to suggest that this study was not performed by professionals. A highly regarded toxicologist carried out the toxicokinetic experiment.

The title contains information about an intestinal integrity test - no such test took place. The structure of the intestinal villi was assessed and properly described. And this is the only value of this manuscript. All elements of toxicological tests are described incorrectly, and the results are unreliable.

A: The present study consisted of two independent experiments. In experiment 1, the broiler chickens were submitted to a toxicokinetic study, where a bolus containing mycotoxins was administered to the broiler chickens. In experiment 2, the broiler chickens were fed a naturally contaminated diet. Also, intestinal morphology was analysed. Therefore, the title of the manuscript was changed to “An algoclay-based decontaminant decreases the exposure to aflatoxin B1, ochratoxin A, and deoxynivalenol in a toxicokinetic model, as well as supports intestinal morphology and decreases liver oxidative stress in broiler chickens fed a diet naturally contaminated with deoxynivalenol”, and instead of integrity, we use now the word morphology. The detailed description of the toxicological study will help the referee to understand the protocol used.

DON is very little susceptible to sorption. Most feed additives used in nutrition are based on the biodegradation of DON by enzymes. No explanation as to why sorption should work so effectively in this experiment. Please explain.

A: Indeed, in vitro data indicates that usually only 20% of DON is adsorbed, and enzymes (if stable during diet pelleting), are more successful in deactivating DON. In the present toxicokinetic experiment, the relative oral bioavailability was decreased by almost 40%, which can be considered a positive result. We have added some references in the Discussion section to clarify this information (Lines 277-286; 352-356).

Round 2

Reviewer 1 Report

Comments and Suggestions for Authors

All my comments have been addressed in the revised manuscript, and I consider that it is suitable for publication in the journal of Toxins.

Comments on the Quality of English Language

Minor editing of English language required

Author Response

A: We would like to acknowledge the present reviewer for the comments. The manuscript was sent again for English language control and the editing was performed.

Reviewer 3 Report

Comments and Suggestions for Authors

Dear Authors, 

Thank You very much for all the additions and explanations introduced to the manuscript. Let me start by making a small remark. I did not claim in my review, that the Authors were forged when providing data. However, I believe that some of the results are unlikely and rarely observed in clinical experience. That's why I asked for a logical explanation of this fact. I also criticized the presentation of toxicological data, especially in lines 424-438. Every toxicologist is interested in the preparation of a biological matrix for analysis, sample extraction and the method of quantitative analysis. Information that the laboratory uses accredited methods is far from sufficient. This is especially true since the laboratory's website describes other research directions than those described in the manuscript (https://www.primoris-lab.com/en/activities/mycotoxins/analysis-methods).

If we provide the results of toxicological analyses, we are obliged to provide the direction of the study and the level of determination (and not "other not detectable"). There is also no explanation for the panel of myctoxins used. Why measure the concentration of beauvercin and Enniatin?

As for the analysis of DON in blood serum - if it is possible at a concentration of 1 mg DON/kg of complete feed, it is even more possible at the higher concentrations that were used in the experiment.

The chapter "DON and DON-3 sulphate in the serum" shows DON concentrations (11.6 µg/ml DON) and DON-3S concentrations ranging from 0.75 to 1.47 ng/ml serum. This destroys the argumentation of laboratory analyses. It seems inappropriate to say "pelleting a diet is a stressful process". The sensitivity of enzymes to high temperature in the granulation process is widely known. That's why we apply enzymes, e.g. by spraying onto cold granules.

Author Response

Reviewer #3

Thank You very much for all the additions and explanations introduced to the manuscript. Let me start by making a small remark. I did not claim in my review, that the Authors were forged when providing data. However, I believe that some of the results are unlikely and rarely observed in clinical experience. That's why I asked for a logical explanation of this fact. I also criticized the presentation of toxicological data, especially in lines 424-438. Every toxicologist is interested in the preparation of a biological matrix for analysis, sample extraction and the method of quantitative analysis. Information that the laboratory uses accredited methods is far from sufficient. This is especially true since the laboratory's website describes other research directions than those described in the manuscript (https://www.primoris-lab.com/en/activities/mycotoxins/analysis-methods).

A: The analysis of the serum was performed and Ghent University, whereas the diets were sent to a private laboratory (Primoris) for a multi-mycotoxin LC-MS/MS analysis. To clarify the description in lines 424-438, the details are now given in lines 404-408 and 438-477.

If we provide the results of toxicological analyses, we are obliged to provide the direction of the study and the level of determination (and not "other not detectable"). There is also no explanation for the panel of myctoxins used. Why measure the concentration of beauvercin and Enniatin?

A: All the details related to the toxicokinetic study are now given. We did not selected specifically the analysis of BEA and ENN, but a multi-mycotoxin analysis, including these ones. Usually, these mycotoxins are present in more than 80% of animal diets. All this information is now given in lines 519-521  and in the Table 7.

As for the analysis of DON in blood serum - if it is possible at a concentration of 1 mg DON/kg of complete feed, it is even more possible at the higher concentrations that were used in the experiment. The chapter "DON and DON-3 sulphate in the serum" shows DON concentrations (11.6 µg/ml DON) and DON-3S concentrations ranging from 0.75 to 1.47 ng/ml serum. This destroys the argumentation of laboratory analyses. 

A: The chickens sampled in the performance trial did not receive a bolus of mycotoxins but were fed ad libitum. We expected DON and DON-3S to be measured. Unfortunately, only the mentioned samples presented DON, and DON-3S levels were variable and low in all samples. Unfortunately, we have no explanation for the absence of measurement in serum. The only difference is that in the performance trial, we used serum, and in the toxicokinetic trial, plasma. However, we cannot state that it is a matrix issue.

It seems inappropriate to say "pelleting a diet is a stressful process". The sensitivity of enzymes to high temperature in the granulation process is widely known. That's why we apply enzymes, e.g. by spraying onto cold granules.

A: Not all feed producers apply enzymes by spraying them onto cold granules, nor do they have the facility to apply liquid enzymes. They usually need injectors and carriers, together with the protection of the system against corrosion, because sometimes they use saltwater to allow the injection of the enzymes. We can share the experience with many feed mills in Europe; most of them still use the dry additive added in the mixer and then pellet together with the diet. This means that sometimes the pellet temperature must be decreased or extra enzymes added, depending on the expected inactivated amount. Decreasing temperatures also pose the risk of affecting pellet quality. We tried to improve the paragraph by explaining the differences with some short sentences.
